# CTCal: Rethinking Text-to-Image Diffusion Models via Cross-Timestep Self-Calibration

## Abstract

Recent advancements in text-to-image synthesis have been largely propelled by diffusion-based models, yet achieving precise alignment between text prompts and generated images remains a persistent challenge. We find that this difficulty arises primarily from the limitations of conventional diffusion loss, which provides only implicit supervision for modeling fine-grained text-image correspondence. In this paper, we introduce Cross-Timestep Self-Calibration (CTCal), founded on the supporting observation that establishing accurate text-image alignment within diffusion models becomes progressively more difficult as the timestep increases. CTCal leverages the reliable text-image alignment (*i.e.*, cross-attention maps) formed at smaller timesteps with less noise to calibrate the representation learning at larger timesteps with more noise, thereby providing explicit supervision during training. We further propose a timestep-aware adaptive weighting to achieve a harmonious integration of CTCal and diffusion loss. CTCal is model-agnostic and can be seamlessly integrated into existing text-to-image diffusion models, encompassing both diffusion-based (*e.g.*, Stable Diffusion 2.1) and flow-based approaches (*e.g.*, Stable Diffusion 3). Extensive experiments on T2I-Compbench++ and GenEval benchmarks demonstrate the effectiveness and generalizability of the proposed CTCal.

## 1 Introduction

Recent advancements in the field of text-to-image synthesis (Nichol et al., 2022; Ramesh et al., 2022; Saharia et al., 2022; Rombach et al., 2022) have been predominantly driven by diffusion-based approaches, particularly those grounded in Denoising Diffusion Probabilistic Models (DDPMs) (Ho et al., 2020; Dhariwal & Nichol, 2021). To further enhance image quality and faithfulness to text prompts, researchers have introduced numerous architectural innovations, including flow-based mechanisms (Albergo & Vanden-Eijnden, 2023; Lipman et al., 2023; Liu et al., 2023; Ma et al., 2024), Diffusion Transformer (DiT) (Peebles & Xie, 2023; Chen et al., 2024b; Li et al., 2024b), Multi-Modal Diffusion Transformer (MM-DiT) (Esser et al., 2024; Labs, 2024), *etc.* Despite these strides, achieving precise and reliable alignment between text prompts and generated images remains an open challenge, especially for complex text prompts, primarily due to limitations in modeling fine-grained text-image correspondence during inference (Hertz et al., 2023; Chefer et al., 2023; Guo et al., 2024) (see Fig. 1 (a)).

Both the cross-attention layer and MM-DiT play the pivotal role in modeling the relationship between text prompts and images, contributing to text-conditioned guidance. These components are typically optimized within existing text-to-image diffusion models utilizing the conventional diffusion loss. However, this implicit approach for learning the text-image correspondence proves to be inadequate for capturing complex correspondences, particularly for larger timesteps with more noise, ultimately impairing the fidelity of the synthesized images.

Existing inference-time optimization methods (Li et al., 2023; Chefer et al., 2023) typically explore the evolution of text-image correspondence (*i.e.*, cross-attention maps) during inference, and suffer from limited generalizability and scalability. In this work, we rethink that from the perspective of the training phase: the challenge of learning the text-image correspondence within text-to-image diffusion models escalates with the progression of timesteps, transitioning from simple to complex scenarios. Empirically, as shown in Fig. 1 (b), cross-attention maps extracted at smaller timesteps

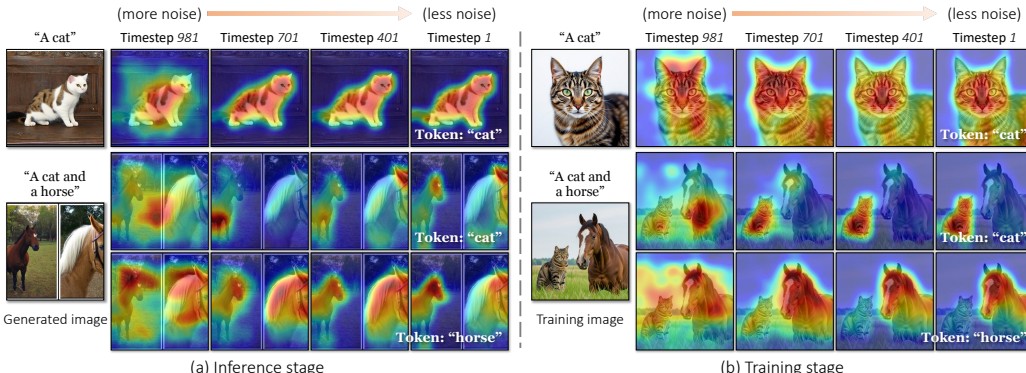

Figure 1: **Investigation on the cross-attention maps. (a) Inference stage.** In line with existing inference-time optimization methods (Hertz et al., 2023; Chefer et al., 2023), we delve into the analysis of cross-attention maps produced during the inference stage of the text-to-image diffusion model. Notably, satisfactory text-image correspondences are established for simple text prompts. Nevertheless, with more intricate text prompts, the prevalent method encounters challenges in precisely mapping the target semantics to the correct spatial position, leading to semantically inconsistent images. **(b) Training stage.** Given the text-image-noise triplet, we gather cross-attention maps at varied timesteps in training mode. A noteworthy finding emerges: cross-attention maps obtained at smaller timesteps exhibit substantially better alignment with the ground-truth image structure and semantics, while this alignment substantially deteriorates at larger timesteps. This suggests that the conventional diffusion loss, which is ubiquitously employed in current training protocols, is effective primarily at smaller timesteps. Moreover, this inability to establish precise alignments at larger timesteps, *i.e.*, initial stage of inference process, constitutes a critical bottleneck, fundamentally constraining the overall fidelity and semantic accuracy of text-to-image generation.

with less noise aligns more accurately with the provided image and corresponds more closely to the semantic distribution in the spatial dimension. This implies that the denoising network can handle the text-image correspondence more effectively under conventional diffusion loss during smaller timesteps. However, this task grows increasingly daunting at larger timesteps.

Drawing from these findings, we introduce Cross-Timestep Self-Calibration (CTCAL), a fine-tuning method that capitalizes on the robust text-image alignment (*i.e.*, cross-attention maps) established at smaller timesteps to calibrate the learning at larger timesteps, achieving explicit self-supervision. Moreover, we propose a part-of-speech-based cross-attention map selection strategy, prioritizing the attention maps corresponding to the noun tokens that contribute most directly to spatial comprehension and eliminating noise interference. We introduce pixel-semantic space joint optimization to augment guidance performance and propose subject response alignment regularization to counteract the potential performance degradation due to unequal subject (noun) response. We achieve a harmonious integration of CTCAL and diffusion loss using a timestep-aware adaptive weighting.

CTCAL is model-agnostic and can be seamlessly integrated into existing text-to-image diffusion models, including both diffusion-based (*e.g.*, Stable Diffusion 2.1) and flow-based approaches (*e.g.*, Stable Diffusion 3). Comprehensive evaluations on T2I-Compbench++ (Huang et al., 2025) and GenEval (Ghosh et al., 2023) benchmarks demonstrate the effectiveness and generalizability.

## 2 PRELIMINARIES

This section presents a brief review of text-to-image diffusion models, cross-attention layer, and multi-modal diffusion transformer, the latter two being instrumental in modeling text-image correspondence and actualizing text-conditioned guidance.

**Text-to-image diffusion models.** Given an image $\mathbf{I}_{\text{real}}$, a text prompt $\mathbf{y}$, a Gaussian noise $\epsilon$, and a timestep $t$, the text-to-image diffusion model $\epsilon_\theta(\cdot)$ is optimized with the following diffusion loss:

$$\mathcal{L}_{\text{diffusion}} = \mathcal{D}\left(\epsilon, \epsilon_\theta\left(\texttt{Add\_Noise}\left(\mathbf{I}_{\text{real}}, \epsilon, t\right), \mathbf{y}, t\right)\right), \tag{1}$$

where $\texttt{Add\_Noise}(\cdot)$ denotes the add noise function and $\mathcal{D}(\cdot)$ is a distance metric, typically implemented as a weighted mean squared error. Although state-of-the-art text-to-image diffusion models

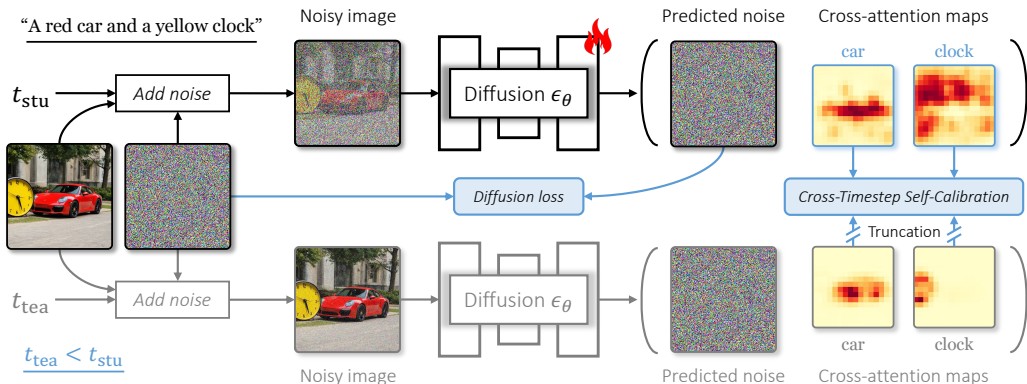

Figure 2: **Illustration of CTCAL.** CTCAL is dedicated to leverage the reliable text-image alignment established at smaller timesteps ($t_{\text{tea}}$) to calibrate the learning process at larger timesteps ($t_{\text{stu}}$). This approach provides explicit supervision for the modeling of text-image correspondence, thereby enhancing the overall performance of text-to-image generation. Notably, the two diffusion models share identical parameters, which is solely for the convenience of presentation.

such as SD 2.1 (Rombach et al., 2022), SD 3 (Esser et al., 2024), and FLUX.1 (Labs, 2024) differ in their specific noise addition and loss formulations, they uniformly adhere to this paradigm.

**Cross-attention layer.** The cross-attention layer is employed to establish the text-image correspondence in classic text-to-image diffusion models (Rombach et al., 2022; Podell et al., 2024). Formally, the feature $f_{\text{image}}(\mathbf{z})$ extracted from the noisy image $\mathbf{z}$ is projected to the query $\mathbf{Q} = \mathcal{Q}(f_{\text{image}}(\mathbf{z}))$, while the text embedding $f_{\text{text}}(\mathbf{y})$ encoded with the provided text prompt $\mathbf{y} = \{\mathbf{y}_1, \mathbf{y}_2, \cdots, \mathbf{y}_n\}$ is projected as the key $\mathbf{K} = \mathcal{K}(f_{\text{text}}(\mathbf{y}))$ and the value $\mathbf{V} = \mathcal{V}(f_{\text{text}}(\mathbf{y}))$, with $\mathcal{Q}(\cdot)$, $\mathcal{K}(\cdot)$, and $\mathcal{V}(\cdot)$ denoting the linear projections. The cross-attention map $\mathbf{A}$ is computed as: $\mathbf{A} = \text{softmax}\left(\frac{\mathbf{Q}\mathbf{K}^T}{\sqrt{d}}\right)$, where $d$ is channel dimension. For ease of representation, we omit the denoising timestep $t$. We denote the cross-attention map that corresponds to the $i$-th text token as $\mathbf{A}_{\mathbf{y}_i}$.

**Multi-modal diffusion transformer (MM-DiT).** Advanced text-to-image diffusion models (Esser et al., 2024; Labs, 2024) introduce the MM-DiT, which diverges from conventional diffusion models by concatenating text and image token embeddings into a unified input sequence. This sequence is then processed by transformer modules that utilize a joint self-attention layer. Formally, MM-DiT is formulated as: $\mathbf{Q} = \text{Concat}(\mathcal{Q}_{\text{image}}(f_{\text{image}}(\mathbf{z})), \mathcal{Q}_{\text{text}}(f_{\text{text}}(\mathbf{y})))$, $\mathbf{K} = \text{Concat}(\mathcal{K}_{\text{image}}(f_{\text{image}}(\mathbf{z})), \mathcal{K}_{\text{text}}(f_{\text{text}}(\mathbf{y})))$, and $\mathbf{V} = \text{Concat}(\mathcal{V}_{\text{image}}(f_{\text{image}}(\mathbf{z})), \mathcal{V}_{\text{text}}(f_{\text{text}}(\mathbf{y})))$, where $\mathcal{Q}_{\text{image}}(\cdot)$, $\mathcal{K}_{\text{image}}(\cdot)$, and $\mathcal{V}_{\text{image}}(\cdot)$ denote the linear projections for image embeddings, and $\mathcal{Q}_{\text{text}}(\cdot)$, $\mathcal{K}_{\text{text}}(\cdot)$, and $\mathcal{V}_{\text{text}}(\cdot)$ denote the linear projections for text embeddings. $\text{Concat}(\cdot)$ is sequence-wise concatenation. The joint self-attention map $\mathbf{A}$ is computed via: $\mathbf{A} = \begin{pmatrix} \mathbf{A}^{\text{II}} & \mathbf{A}^{\text{IT}} \\ \mathbf{A}^{\text{TI}} & \mathbf{A}^{\text{TT}} \end{pmatrix} = \text{softmax}\left(\frac{\mathbf{Q}\mathbf{K}^T}{\sqrt{d}}\right)$, where $d$ is channel dimension. For simplicity, we omit the denoising timestep $t$. In this work, we focus on $\mathbf{A}^{\text{IT}}$, where we denote the cross-attention map that corresponds to the $i$-th text token as $\mathbf{A}_{\mathbf{y}_i}^{\text{IT}}$.

## 3 APPROACH

The core innovation of our approach is Cross-Timestep Self-Calibration (CTCAL), which leverages reliable text-image alignments learned at small timesteps to calibrate the learning at larger timesteps. This section is organized as follows. Sec. 3.1 provides an overview of the training paradigm. Sec. 3.2 details a comprehensive description of the CTCAL method. Sec. 3.3 outlines our training strategy.

### 3.1 OVERVIEW

Fig. 2 illustrates our proposed training paradigm, which deviates from the conventional training approach. Given a real image $\mathbf{I}_{\text{real}}$, a text prompt $\mathbf{y}$, and a Gaussian noise $\epsilon$, we sample two distinct timesteps, referred to as $t_{\text{stu}}$ and $t_{\text{tea}}$, with $t_{\text{tea}} < t_{\text{stu}}$. Besides predicting the corresponding noises

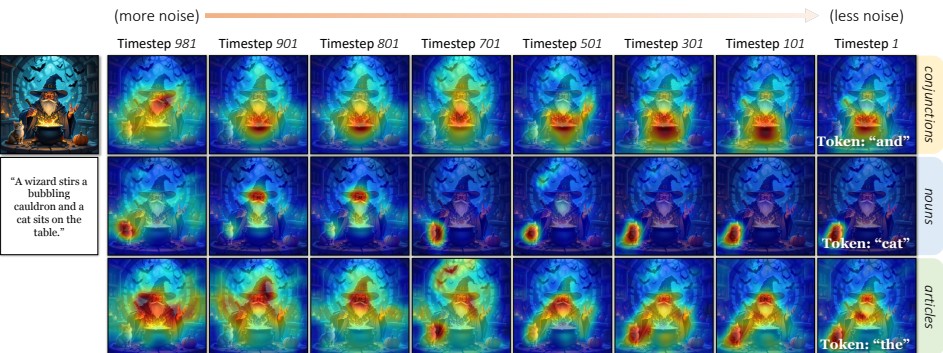

Figure 3: **Investigation on cross-attention maps categorized by part-of-speech.** Cross-ttention maps for noun tokens (*e.g.*, "cat") typically encode clear spatial semantic information, while those for articles (*e.g.*, "the") and conjunctions (*e.g.*, "and") seemingly lack a significant conveyance.

$\epsilon_{\text{stu}}$ and $\epsilon_{\text{tea}}$, we also extract and store the cross-attention maps $\mathbf{A}_{\text{stu}}$ and $\mathbf{A}_{\text{tea}}$, computed during the forward process of the denoising network. Notably, both $\mathbf{A}_{\text{stu}}$ and $\mathbf{A}_{\text{tea}}$ are extracted from the same diffusion model, which is currently being fine-tuned. Unlike constructing a separate and fixed pre-trained model for extracting $\mathbf{A}_{\text{tea}}$, our design allows $\mathbf{A}_{\text{tea}}$ to benefit from the learning on newly introduced high-quality data. The aggregated map $\mathbf{A}_{\text{stu/tea}} \in \mathbb{R}^{H \times W \times n}$ consists of $n$ spatial attention maps, each associated with a token of the text prompt. More details on the workflow for processing cross-attention maps are provided in the supplementary material.

Notably, we restrict the optimization to the denoising network associated with timestep $t_{\text{stu}}$, and truncate the gradient of $\mathbf{A}_{\text{tea}}$. Furthermore, we leverage the cross-attention maps $\mathbf{A}_{\text{tea}}$ derived from smaller timestep $t_{\text{tea}}$ as a guide for learning the cross-attention maps $\mathbf{A}_{\text{stu}}$ from larger timestep $t_{\text{stu}}$. This approach explicitly transfers knowledge about text-image correspondence, which is more accurately captured at smaller timesteps, to enhance the learning at larger timesteps. Consequently, the optimization objective is redefined as:

$$\mathcal{L} = \mathcal{L}_{\text{diffusion}} + \mathcal{L}_{\text{CTCAL}} = \mathcal{D}\left(\epsilon, \epsilon_\theta\left(\texttt{Add\_Noise}\left(\mathbf{I}_{\text{real}}, \epsilon, t_{\text{stu}}\right), \mathbf{y}, t_{\text{stu}}\right)\right) + \mathcal{D}\left(\mathbf{A}_{\text{stu}}, \mathbf{A}_{\text{tea}}\right). \quad (2)$$

## 3.2 CTCAL

This section provide a detailed explanation of Cross-Timestep Self-Calibration (CTCAL), which consists of the following three carefully designed components.

**Part-of-speech-based cross-attention map selection strategy.** Given an aggregated cross-attention map $\mathbf{A}_{\text{stu/tea}} \in \mathbb{R}^{H \times W \times n}$, consisting of $n$ spatial attention maps. However, as shown in Fig. 3, not all tokens yield attention maps that encapsulate meaningful spatial semantic information. For example, tokens representing articles (*e.g.*, "the") and conjunctions (*e.g.*, "and") may not convey meaningful spatial semantics. Overemphasis on them could potentially degrade the performance.

To rectify this, we propose a part-of-speech-based cross-attention map selection strategy that only extracts and utilizes the attention maps associated with tokens likely to convey significant spatial semantics, specifically, nouns (denoting objects or entities). We reformulate $\mathcal{L}_{\text{CTCAL}}$ as follows:

$$\mathcal{L}_{\text{CTCAL}} = \frac{1}{N_{\text{noun}}} \sum_{\mathbf{y}_i \in \mathcal{Y}_{\text{noun}}} \mathcal{D}\left(\mathbf{A}_{\text{stu}, \mathbf{y}_i}, \mathbf{A}_{\text{tea}, \mathbf{y}_i}\right), \quad (3)$$

where $\mathcal{Y}_{\text{noun}}$ denotes the set of noun tokens, and $N_{\text{noun}}$ is the number of noun tokens. By restricting the selection of attention maps to this subset, CTCAL prioritizes tokens that contribute most directly to spatial understanding.

**Pixel-semantic space joint optimization.** To achieve alignment between $\mathbf{A}_{\text{stu}}$ and $\mathbf{A}_{\text{tea}}$, we propose a joint optimization paradigm that simultaneously considers both pixel-level and semantic-level representations. Empirical evidence substantiates the superior performance of this methodology as against an exclusive emphasis on either of the two constituents. We redefine the $\mathcal{L}_{\text{CTCAL}}$ as follows:

$$\mathcal{L}_{\text{CTCAL}} = \frac{1}{N_{\text{noun}}} \sum_{\mathbf{y}_i \in \mathcal{Y}_{\text{noun}}} \lambda_1 \mathcal{D}\left(\mathbf{A}_{\text{stu}, \mathbf{y}_i}, \mathbf{A}_{\text{tea}, \mathbf{y}_i}\right) + \lambda_2 \mathcal{D}\left(f_{\text{attn}}\left(\mathbf{A}_{\text{stu}, \mathbf{y}_i}\right), f_{\text{attn}}\left(\mathbf{A}_{\text{tea}, \mathbf{y}_i}\right)\right), \quad (4)$$

where $f_{\text{attn}}(\cdot)$ denotes the feature encoder that projects attention maps to their respective semantic representations. A notable concern is the potential overfitting of $f_{\text{attn}}(\cdot)$ during training, which may instigate mode collapse, causing $f_{\text{attn}}(\cdot)$ to project all attention maps to identical feature encodings.

To mitigate this risk, we devise a lightweight autoencoder, composed of an encoder $f_{\text{attn}}^{\text{enc}}(\cdot)$ and a decoder $f_{\text{attn}}^{\text{dec}}(\cdot)$. We apply a reconstruction proxy task as a preventive measure against overfitting:

$$\mathcal{L}_{\text{CTCAL}} = \frac{1}{N_{\text{noun}}} \sum_{\mathbf{y}_i \in \mathcal{Y}_{\text{noun}}} \lambda_1 \mathcal{D}\left(\mathbf{A}_{\text{stu},\mathbf{y}_i}, \mathbf{A}_{\text{tea},\mathbf{y}_i}\right) + \lambda_2 \mathcal{D}\left(f_{\text{attn}}^{\text{enc}}\left(\mathbf{A}_{\text{stu},\mathbf{y}_i}\right), f_{\text{attn}}^{\text{enc}}\left(\mathbf{A}_{\text{tea},\mathbf{y}_i}\right)\right)$$
$$+ \lambda_3 \mathcal{D}\left(f_{\text{attn}}^{\text{dec}}\left(f_{\text{attn}}^{\text{enc}}\left(\mathbf{A}_{\text{tea},\mathbf{y}_i}\right)\right), \mathbf{A}_{\text{tea},\mathbf{y}_i}\right). \tag{5}$$

The detailed architecture of the proposed autoencoder is presented in the supplementary material.

**Subject response alignment regularization.** CTCAL focuses on spatial alignment but may suffer from the imbalanced cross-attention responses among subjects, *i.e.*, subjects with higher responses may overshadow those with lower responses, resulting in the latter being ineffectively rendered in the generated image. Therefore, we introduce subject response alignment regularization, which aligns the cross-attention responses of all subjects to that of the subject with the highest response:

$$\mathcal{R}_{\text{subject}} = \frac{1}{N_{\text{noun}}} \sum_{\mathbf{y}_i \in \mathcal{Y}_{\text{noun}}} \text{ReLU}\left(\mathcal{S}_{\text{attn}} - \max\left(\mathbf{A}_{\text{stu},\mathbf{y}_i}\right) - \tau\right), \tag{6}$$

where $\mathcal{S}_{\text{attn}} = \max_{\mathbf{y}_i \in \mathcal{Y}_{\text{noun}}} \max\left(\mathbf{A}_{\text{stu},\mathbf{y}_i}\right)$. Following (Chefer et al., 2023), we use $\max\left(\mathbf{A}_{\text{stu},\mathbf{y}_i}\right)$ to characterize the attention response level of subject token $\mathbf{y}_i$. $\tau$ denotes the threshold, and $\text{ReLU}(\cdot)$ ensures that only pairs with differences exceeding $\tau$ contribute to the loss. This design ensures the comparability of responses across different subjects, while effectively preventing the attention responses from increasing unconstrainedly during training.

**CTCAL.** In summary, $\mathcal{L}_{\text{CTCAL}}$ is ultimately represented as:

$$\mathcal{L}_{\text{CTCAL}} = \frac{1}{N_{\text{noun}}} \sum_{\mathbf{y}_i \in \mathcal{Y}_{\text{noun}}} \lambda_1 \underbrace{\mathcal{D}\left(\mathbf{A}_{\text{stu},\mathbf{y}_i}, \mathbf{A}_{\text{tea},\mathbf{y}_i}\right)}_{\text{Pixel-level loss}} + \lambda_2 \underbrace{\mathcal{D}\left(f_{\text{attn}}^{\text{enc}}\left(\mathbf{A}_{\text{stu},\mathbf{y}_i}\right), f_{\text{attn}}^{\text{enc}}\left(\mathbf{A}_{\text{tea},\mathbf{y}_i}\right)\right)}_{\text{Semantic-level loss}}$$
$$+ \lambda_3 \underbrace{\mathcal{D}\left(f_{\text{attn}}^{\text{dec}}\left(\mathbf{A}_{\text{tea},\mathbf{y}_i}\right), \mathbf{A}_{\text{tea},\mathbf{y}_i}\right)}_{\text{Reconstruction proxy task}} + \lambda_4 \underbrace{\mathcal{R}_{\text{subject}}}_{\text{Regularization}}, \tag{7}$$

where $\lambda_1$, $\lambda_2$, $\lambda_3$, and $\lambda_4$ are the tradeoff parameters. Further details on the tradeoff parameters settings are provided in the supplementary material.

### 3.3 TRAINING STRATEGY

**Training timestep sampling strategy.** For $t_{\text{stu}}$, we strictly adhere to the inherent timestep sampling protocol that has been established by the text-to-image diffusion models during the training stage. For $t_{\text{tea}}$, we empirically set $t_{\text{tea}} = 0$ for classical text-to-image diffusion models (*e.g.*, Stable Diffusion 2.1). This particular selection corresponds to the regime with minimal noise.

It is critical to highlight that, in contrast to classical methods that rely on uniform timestep sampling, contemporary cutting-edge models (*e.g.*, Stable Diffusion 3) have integrated non-uniform timestep samplers, such as the logit-normal sampler. Consequently, a reevaluation of timestep priority based on the sampling distribution becomes crucial for identifying $t_{\text{tea}}$. Naively setting $t_{\text{tea}} = 0$ may thus degrade performance. Further discussion and implementation details specific to Stable Diffusion 3 are provided in the supplementary material.

**Timestep-aware adaptive weighting.** To augment the potency of $\mathcal{L}_{\text{CTCAL}}$, we introduce a timestep-aware adaptive weighting scheme. Specifically, during the initial stages (*i.e.*, with less noise) of the diffusion process, the diffusion loss predominantly governs the alignment between textual and visual modalities, rendering a lower contribution from $\mathcal{L}_{\text{CTCAL}}$. In contrast, at larger timesteps (*i.e.*, with more noise), the model relies more heavily on $\mathcal{L}_{\text{CTCAL}}$.

We formalize this intuition using a simple yet effective linear weighting function that scales the influence of $\mathcal{L}_{\text{CTCAL}}$ according to the current diffusion timestep:

$$\mathcal{L} = \mathcal{L}_{\text{diffusion}} + \lambda_t \mathcal{L}_{\text{CTCAL}}, \quad \text{where } \lambda_t = \frac{t_{\text{stu}}}{T_{\text{train}}}, \tag{8}$$

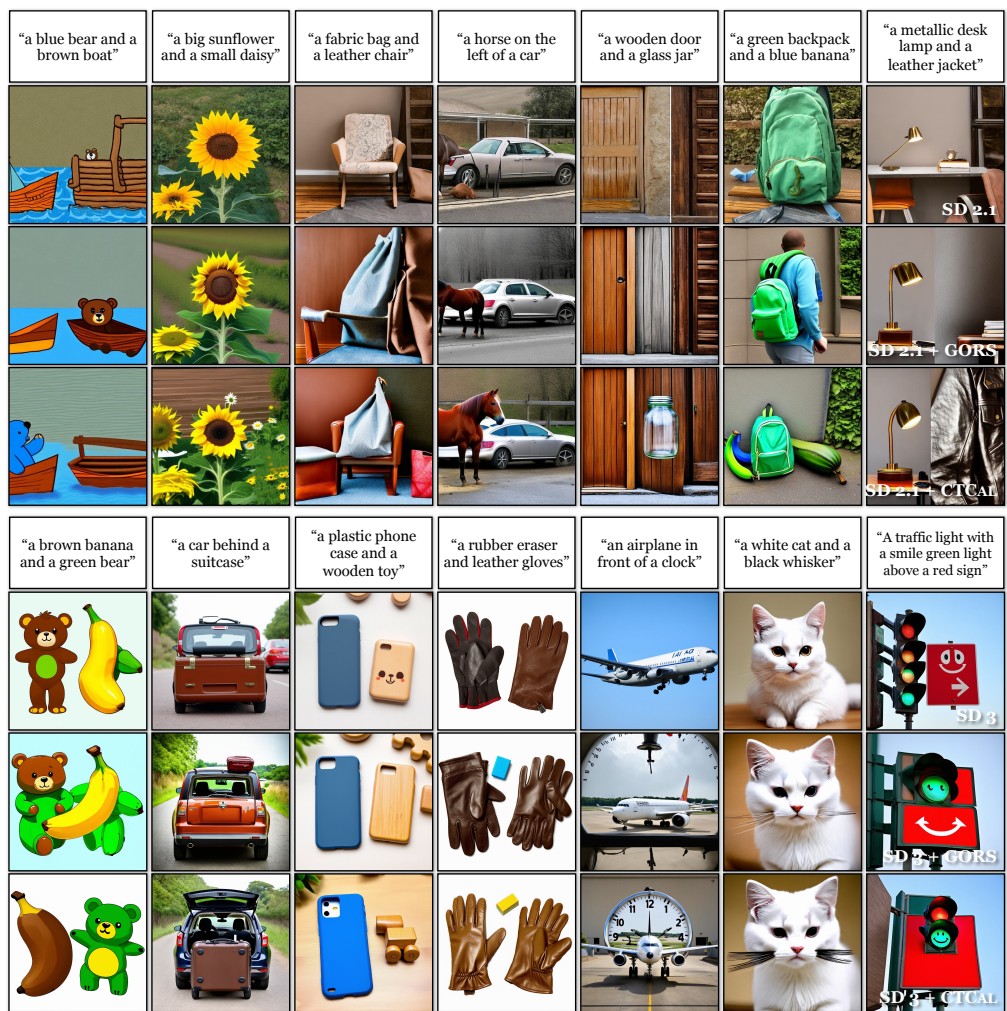

Figure 4: **Qualitative comparison on SD 2.1 and SD 3.** CTCAL demonstrates a marked improvement in the fine-grained alignment of generated images with the corresponding text prompts. Each image is generated with the same text prompt and random seed for all methods.

where $t_{stu}$ is the current timestep and $T_{train}$ is the total number of diffusion steps during training. Thus, $\lambda_t$ increases linearly with $t_{stu}$, assigning greater emphasis to $\mathcal{L}_{CTCAL}$ as the process advances. This adaptive scheme enables the model to balance both objectives throughout training, facilitating stable convergence and improved text-image alignment.

## 4 EXPERIMENTS

### 4.1 EXPERIMENTAL SETTINGS

**Implementation details.** CTCAL is a model-agnostic training paradigm that can be seamlessly incorporated into prevailing text-to-image diffusion frameworks. To validate the efficacy and generalizability of CTCAL, we integrate it with two highly recognized diffusion models: Stable Diffusion 2.1 (SD 2.1) (Rombach et al., 2022) and Stable Diffusion 3 (SD 3) (Esser et al., 2024). CTCAL is implemented within the Diffusers codebase, employing Low-Rank Adaptation (LoRA) to fine-tune both the self-attention layers of the text encoder and the attention layers of the denoising network. We use *Stanza* for part-of-speech analysis to extract nouns from the given text prompts. $\mathcal{D}(\cdot)$ is implemented as the mean squared error loss function for CTCAL. We conduct a comprehensive evaluation of CTCAL utilizing two widely recognized benchmarks: T2I-CompBench++ (Huang et al., 2025) and GenEval (Ghosh et al., 2023). More parameter setting, training and evaluation details are provided in the supplementary material.

Table 1: **Objective evaluation on T2I-CompBench++.** Top: conventional text-to-image diffusion models, inference-time optimization approaches, supervised fine-tuning methods, and our method. Bottom: advanced text-to-image diffusion models, supervised fine-tuning methods, and our method. CTCAL exhibits the superior performance in attribute binding, object relationships, counting, and complex compositions, highlighting the advanced capability for compositional generation.

| Methods | Color B-VQA | Shape B-VQA | Texture B-VQA | 2D-Spatial UniDet | 3D-Spatial UniDet | Numeracy UniDet | Non-Spatial Share-CoT | Complex 3-in-1 |
|---|---|---|---|---|---|---|---|---|
| SD 1.4 (Rombach et al., 2022) | 0.3765 | 0.3576 | 0.4156 | 0.1246 | 0.3030 | 0.4456 | 0.7487 | 0.3080 |
| SD 2.1 (Rombach et al., 2022) | 0.5065 | 0.4221 | 0.4922 | 0.1342 | 0.3230 | 0.4582 | 0.7567 | 0.3386 |
| SD 2.1 + CD (Liu et al., 2022) | 0.4063 | 0.3299 | 0.3645 | 0.0800 | 0.2847 | 0.4272 | 0.6927 | 0.2898 |
| SD 2.1 + SD (Feng et al., 2023) | 0.4990 | 0.4218 | 0.4900 | 0.1386 | 0.3224 | 0.4557 | 0.7560 | 0.3355 |
| SD 2.1 + AE (Chefer et al., 2023) | 0.6400 | 0.4517 | 0.5963 | 0.1455 | 0.3222 | 0.4773 | 0.7593 | 0.3401 |
| SD 2.1 + GORS[1] | 0.6603 | 0.4785 | 0.6287 | 0.1815 | 0.3572 | 0.4830 | 0.7637 | 0.3328 |
| SD 2.1 + GORS[2] | 0.6426 | 0.4864 | 0.6319 | 0.1775 | 0.3475 | 0.4856 | 0.7621 | 0.3371 |
| SD 2.1 + CTCAL | **0.7233** | **0.5149** | **0.6754** | **0.2142** | **0.3862** | **0.5084** | **0.7723** | **0.3403** |
| SD XL (Podell et al., 2024) | 0.5879 | 0.4687 | 0.5299 | 0.2133 | 0.3566 | 0.4991 | 0.7673 | 0.3237 |
| Pixart-$\alpha$-ft (Chen et al., 2024b) | 0.6690 | 0.4927 | 0.6477 | 0.2064 | 0.3901 | 0.5032 | 0.7747 | 0.3433 |
| DALL-E 3 (Betker et al., 2023) | 0.7785 | 0.6205 | 0.7036 | 0.2865 | 0.3744 | 0.5926 | 0.7853 | 0.3773 |
| FLUX-schnell (Labs, 2024) | 0.7407 | 0.5718 | 0.6922 | 0.2863 | 0.3866 | 0.6185 | 0.7809 | 0.3703 |
| SD 3 (2B) (Esser et al., 2024) | 0.8132 | 0.5885 | 0.7334 | 0.3200 | 0.4084 | 0.6174 | 0.7782 | 0.3771 |
| SD 3 (2B) + CORS[2] | 0.8236 | 0.5833 | 0.7398 | 0.3232 | 0.4033 | 0.6280 | 0.7708 | 0.3739 |
| SD 3 (2B) + CTCAL | **0.8443** | **0.5968** | **0.7581** | **0.3476** | **0.4117** | **0.6292** | **0.7867** | **0.3814** |

[1] The results are sourced from the original paper (Huang et al., 2025).
[2] The results are derived from our reimplementation using the text-image dataset we constructed.

**Datasets.** Current mainstream text-to-image generation models (Betker et al., 2023; Podell et al., 2024; Li et al., 2024b; Chen et al., 2024b; Labs, 2024; Esser et al., 2024; Xie et al., 2025) are predominantly trained on proprietary datasets, resulting in a paucity of open-source, high-quality text-image pair datasets within the research community. To address this limitation, we adopt the dataset construction method proposed by (Huang et al., 2025), which utilizes a reward-driven sample selection strategy to curate training dataset. Specifically, we utilize the text prompt dataset from (Huang et al., 2025), which comprises 700 prompts per category. For each prompt, we generate $k$ images using the target text-to-image diffusion model, thereby forming a set of candidate text-image pairs. Each candidate pair is then evaluated using the scoring metric introduced in (Huang et al., 2025). We subsequently select the top-$n$ pairs with the highest scores from each candidate set to fine-tune the diffusion model. In our experiments, for each category in (Huang et al., 2025), we set $k = 100$, $n = 10,000$ for SD 2.1, and $k = 30$, $n = 10,000$ for SD 3.

## 4.2 QUALITATIVE COMPARISON

Fig. 4 present a comparative analysis of our method against supervised fine-tuning approach (*i.e.*, GORS) using identical text prompts and random seeds on Stable Diffusion 2.1 (SD 2.1) and Stable Diffusion 3 (SD 3). GORS leverages synthesized text-image data meticulously selected based on reward functions as detailed in Sec. 4.1, and adopts the standard diffusion loss for model fine-tuning. Building on this, our approach incorporates CTCAL.

As illustrated in Fig 4, SD 2.1 struggles with compositional text-to-image synthesis. While GORS demonstrates improvements, it still exhibits limitations in accurately rendering uncommon concepts. For instance, GORS enhances the depiction of "blue" but fails to render a "blue banana." In contrast, our method successfully synthesizes such challenging compositions. Fig. 4 shows that SD 3, benefiting from extensive high-quality datasets and advanced architectures, already achieve strong performance on text-guided image generation. Nonetheless, CTCAL further enhances performance beyond this baseline.

## 4.3 QUANTITATIVE COMPARISON

**Objective evaluation.** Table 1 presents quantitative results on T2I-CompBench++ (Huang et al., 2025). CTCAL demonstrates substantial improvements over existing text-to-image diffusion models

Table 2: **Objective evaluation on GenEval.** CTCAL improves performance across all categories.

| Methods | Overall | Single object | Two object | Counting | Colors | Position | Color attribution |
|---|---|---|---|---|---|---|---|
| SD 1.5 (Rombach et al., 2022) | 0.43 | 0.97 | 0.38 | 0.35 | 0.76 | 0.04 | 0.06 |
| SD 2.1 (Rombach et al., 2022) | 0.50 | 0.98 | 0.51 | 0.44 | 0.85 | 0.07 | 0.17 |
| SD 3 (2B) (Esser et al., 2024) | 0.62 | 0.98 | 0.74 | 0.63 | 0.67 | 0.34 | 0.36 |
| SD 3 (2B) + CTCAL | **0.69** | **0.99** | **0.85** | **0.70** | **0.79** | **0.38** | **0.42** |

in attribute binding, object relationships, counting, and complex compositions, including diffusion-based (SD 2.1) and flow-based method (SD 3). Furthermore, CTCAL outperforms inference-time optimization and supervised fine-tuning methods, confirming the effectiveness and generalizability.

Notably, noun-token-based CTCAL still enhances performance on the dimensions of action and positional relationship. This is primarily attributed to the supervision of accurate subject rendering, which also improves the ability to understand and learn from training images. Furthermore, rendering the subject at the correct position partially integrates positional and action information. The manifestation of both positional and action information depends on the subject, which serves as the foundation for positional relationships.

To mitigate potential biases introduced by employing evaluation metrics as rewards during dataset construction, we further report cross-benchmark validation results in Table 2. Unlike the protocol in (Huang et al., 2025), which fine-tunes LoRA parameters for specific categories, we aggregate text-image pairs of all categories (80,000 pairs) for joint fine-tuning. As shown in Table 2, evaluation on GenEval (Ghosh et al., 2023) indicates that CTCAL consistently improves performance across all categories, further substantiating the robustness of our approach.

Table 3: **User study.**

| Methods | User study |
|---|---|
| SD 2.1 | 4.17% |
| + GORS | 19.17% |
| + CTCAL | **76.67%** |
| SD 3 | 24.17 |
| + GORS | 21.67 |
| + CTCAL | **54.17** |

**User study.** A subjective user study is conducted with 12 volunteers, 6 of whom have expertise in image processing. Participants are asked to select the most visually appealing and semantically faithful images, with 10 questions per participant. We record the voting results and present the statistics in Table 3. Our method performs favorably against the other methods.

## 4.4 Ablation Study

We perform the ablation study on the Color and 2D-Spatial categories of T2I-CompBench++ to systematically evaluate the effectiveness of our design. We define as follows: (A) denotes the naive constraint of $\mathbf{A}_{\text{tea}}$ and $\mathbf{A}_{\text{stu}}$, (B) introduces a part-of-speech-based cross-attention map selection strategy based on (A), (C) introduces pixel-semantic space joint optimization based on (B), (D) introduces subject response alignment regularization based on (C), and

Table 4: **Ablation study** on Stable Diffusion 2.1.

| Methods | Color B-VQA | 2D-Spatial UniDet |
|---|---|---|
| SD 2.1 | 0.5065 | 0.1342 |
| + GORS (baseline) | 0.6426 | 0.1775 |
| + CTCAL (A) | 0.6286 (-2.18%) | 0.1693 (-4.62%) |
| + CTCAL (B) | 0.6897 (+7.33%) | 0.1972 (+11.10%) |
| + CTCAL (C) | 0.6992 (+8.81%) | 0.2021 (+13.86%) |
| + CTCAL (D) | 0.7148 (+11.24%) | 0.2095 (+18.03%) |
| + CTCAL (E) | **0.7233 (+12.56%)** | **0.2142 (+20.68%)** |

(E) introduces timestep-aware adaptive weighting based on (D). (E) is the final version of CTCAL

As shown in Table 4, (A) even decreases performance due to considering the attention maps that do not contain spatial semantic information. With the priority given to noun tokens by the part-of-speech-based cross-attention map selection strategy, (B) demonstrates its effectiveness, significantly enhancing performance. The Pixel-semantic space joint optimization, Subject response alignment regularization, and Timestep-aware adaptive weighting further optimize the performance of CTCAL.

## 4.5 More Results

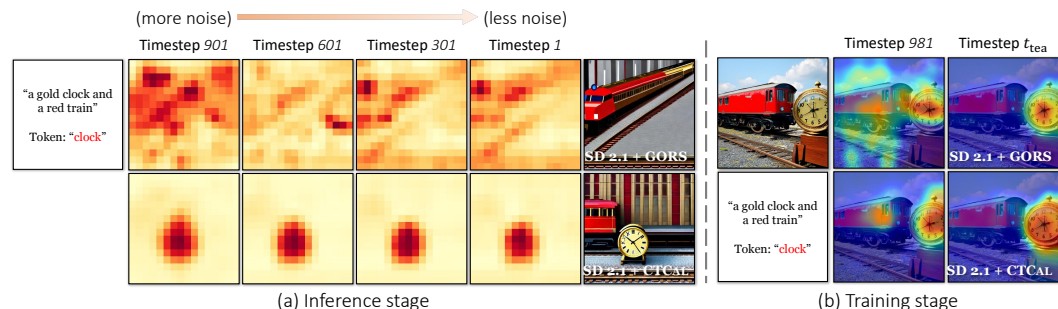

Figure 5: **Visualization of cross-attention maps** extracted from the fine-tuned models.

**More results on $t_{\text{tea}}$.** As mentioned in Sec. 3.2, CTCAL requires $t_{\text{tea}} < t_{\text{stu}}$. For conventional diffusion models (*e.g.*, Stable Diffusion 2.1), we empirically set $t_{\text{tea}}$ to a fixed value of 0. This specific choice corresponds to the scenario with the least noise, representing the most favorable timestep for learning text-image correspondences. To further investigate this, we conducted additional experiments. Instead of setting a fixed $t_{\text{tea}} = 0$, we randomly sample $t_{\text{tea}}$ for a given $t_{\text{stu}}$ such that $0 \leq t_{\text{tea}} < t_{\text{stu}}$. The quantitative evaluation presented in Table 5 shows that while the random sampling approach can still enhance the performance, setting $t_{\text{tea}} = 0$ remains the superior choice.

Table 5: **Objective evaluation on $t_{\text{tea}}$.**

| Methods | Color B-VQA | 2D-Spatial UniDet |
|---|---|---|
| SD 2.1 | 0.5065 | 0.1342 |
| + CTCAL ($0 \leq t_{\text{tea}} < t_{\text{stu}}$) | 0.7028 | 0.2029 |
| + CTCAL (Ours) | **0.7233** | **0.2142** |

**More results on the part-of-speech-based cross-attention map selection strategy.** As discussed in Sec. 3.2, cross-attention maps corresponding to noun tokens generally encapsulate clear spatial-semantic information. Therefore, CTCAL prioritizes the attention maps corresponding to noun tokens. Notably, adjectives, especially those modifying nouns, also demonstrate accurate spatial correspondence. Accordingly, we further discuss tokens with adjectival properties. Table 6 presents additional quantitative results on the Color and Texture categories of the T2I-CompBench++ benchmark. These experiments focus on adjective-noun pairs to more rigorously assess attribute rendering accuracy. The results indicate that incorporating alignment on adjective tokens leads to measurable improvements in performance.

Table 6: **Objective evaluation on *adj.*.**

| Methods | Color B-VQA | Texture B-VQA |
|---|---|---|
| SD 2.1 | 0.5065 | 0.4922 |
| + GORS (baseline) | 0.6426 | 0.6319 |
| + CTCAL | 0.7233 | 0.6754 |
| + CTCAL + adj. token | **0.7328** | **0.6877** |

**Visualizations of cross-attention maps.** Fig. 5 presents visualizations of cross-attention maps generated by the fine-tuned models, depicted separately for (a) inference and (b) training modes. During inference, our method demonstrates accurate and reasonable attention allocation, leading to semantically consistent outputs. Furthermore, compared to GORS, the cross-attention maps derived from CTCAL at later timesteps exhibit greater consistency with those at smaller timesteps. This observation underscores the efficacy of our approach and provides empirical support for our insights.

## 5  CONCLUSION

This study addresses the persistent challenge of precise text-image alignment in text-to-image diffusion models by introducing Cross-Timestep Self-calibration (CTCAL). Through a rigorous analysis, we demonstrate that alignment difficulties intensify with increasing diffusion timesteps, underscoring the limitations of conventional diffusion loss. CTCAL mitigates this issue by explicitly calibrating the learning at larger timesteps with more noise using the robust text-image alignment established at smaller timesteps with less noise, supplemented by a timestep-aware adaptive weighting mechanism for seamless integration with standard diffusion losses. CTCAL is model-agnostic and readily adaptable to a wide array of diffusion-based and flow-based architectures. Extensive evaluation on established benchmarks substantiates the efficacy and generalizability of CTCAL, marking a significant advancement toward more accurate and reliable text-to-image generation.

## 6 ETHICS STATEMENT

We uphold the highest ethical standards in our research, which includes complying with legal frameworks, respecting privacy rights, and encouraging the generation of positive content. Text-to-image models have a wide range of applications across diverse scenarios. Although these models may be misused for harmful purposes, it is essential to employ safety checkers or filters during actual deployment to prevent the generation of NSFW content. In this work, we rely on existing pre-trained models and therefore inherit their inherent biases. However, our training framework is highly flexible and, by utilizing customized, safe, and trustworthy datasets for fine-tuning, can effectively mitigate the potential harms associated with current image generation models.

## 7 REPRODUCIBILITY STATEMENT

We provide complete implementation details for the training and evaluation of the proposed method in the main paper (Sec. 4.1) and supplementary material (Appendix C) to enhance reproducibility. The code and weights will be open-sourced upon acceptance of this paper to facilitate further research and exploration.

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

APPENDIX

This appendix is structured as follows:

- In Appendix A, we provide a discussion of related work.
- In Appendix B, we provide additional implementation details regarding CTCAL, including the processing workflow for cross-attention maps, the architecture of the proposed lightweight autoencoder, and the training timestep sampling strategy for SD 3.
- In Appendix C, we provide more details of our experimental settings, including tradeoff parameters and benchmarks.
- In Appendix D, we provide additional results and analysis, including more results on diversity evaluation, more results on image quality evaluation, more application on inference acceleration, and additional qualitative comparisons.
- In Appendix E, we provide the limitations of CTCAL.
- In Appendix F, we provide details on the usage of Large Language Models.

## A    RELATED WORK

Text-to-image synthesis aims to generate visually realistic images that accurately reflect input text prompts. Early advances in this field were predominantly driven by Generative Adversarial Networks (GANs) (Zhang et al., 2017; Xu et al., 2018; Zhu et al., 2019; Zhang et al., 2021; Tao et al., 2022) and Autoregressive Models (ARs) (Ramesh et al., 2021; Ding et al., 2021; Yu et al., 2022; Chang et al., 2023). More recently, Diffusion Models (DMs) (Ho et al., 2020; Dhariwal & Nichol, 2021) have emerged as the dominant paradigm, demonstrating superior capabilities in generating high-fidelity and semantically coherent images.

Building upon the foundational framework of Denoising Diffusion Probabilistic Models (DDPMs), diffusion-based approaches have become the cornerstone of contemporary text-to-image synthesis. Representative models include GLIDE (Nichol et al., 2022), which extends diffusion models for text-guided image generation and editing, DALL-E 2 (Ramesh et al., 2022), which leverages the CLIP embedding space for improved text-image generation, Imagen (Saharia et al., 2022), which employs cascaded diffusion for high-resolution synthesis, and Stable Diffusion (SD) (Rombach et al., 2022), which adopts latent diffusion for computational efficiency.

To further advance the quality and fidelity of generated images, researchers have proposed a range of architectural innovations. These include the integration of flow-based method (Albergo & Vanden-Eijnden, 2023; Lipman et al., 2023; Liu et al., 2023; Ma et al., 2024), the development of Diffusion Transformers (DiT) (Peebles & Xie, 2023; Chen et al., 2024b; Li et al., 2024b), and the introduction of Multi-Modal Diffusion Transformer (MM-DiT) (Esser et al., 2024; Labs, 2024). Notable recent advancements in this direction include PixArt-$\alpha$ (Chen et al., 2024b), HunyuanDiT (Li et al., 2024b), FLUX.1 (Labs, 2024), Stable Diffusion 3 (Esser et al., 2024), and HiDream-I1.

Beyond architectural improvements, fine-tuning strategies have been extensively explored to further enhance text-to-image diffusion models. Data augmentation-based methods (Huang et al., 2023; Wu et al., 2023; Lee et al., 2023; Dong et al., 2023; Dai et al., 2023; Betker et al., 2023; Segalis et al., 2023) focus on modifying the training data distribution to improve both visual fidelity and textual alignment. Backpropagation-based approaches (Xu et al., 2023; Clark et al., 2024; Prabhudesai et al., 2023; Wu et al., 2024) employ differentiable reward functions, enabling end-to-end optimization via gradient descent. Reinforcement learning-based methods (Black et al., 2023; Deng et al., 2024; Chen et al., 2024a; Fan et al., 2023; Zhang et al., 2024) incorporate Reinforcement Learning from Human Feedback (RLHF), allowing models to iteratively refine their outputs based on reward signals. Furthermore, Direct Preference Optimization (DPO) based approaches (Wallace et al., 2024; Yang et al., 2024a; Liang et al., 2024; Yang et al., 2024b; Li et al., 2024a; Zhu et al., 2025) circumvent the complexities of explicit reward modeling by directly optimizing for user preferences.

Despite these advances, existing methods largely overlook the explicit supervision of text-image correspondence learning, leading to suboptimal results. In this work, we address this gap by lever-

aging a fundamental characteristic of text-to-image diffusion models, utilizing the robust text-image alignment established at smaller timesteps to supervise and calibrate the learning at larger timesteps.

# B    CTCAL

**Processing workflow for cross-attention maps.** Given an input triplet comprising a real image, a corresponding text prompt, and Gaussian noise, we first sample a specific timestep and subsequently extract the associated cross-attention maps generated during the forward process of the denoising network. These maps are subsequently averaged across all layers and attention heads to yield the final cross-attention map. The aggregated map $\mathbf{A} \in \mathbb{R}^{H \times W \times n}$ consists of $n$ spatial attention maps, each corresponding to a token in the text prompt.

For Stable Diffusion 2.1, the attention maps are extracted at a spatial resolution of $16 \times 16$ pixels. Consistent with the methodology described in (Chefer et al., 2023), we apply a Gaussian filter with a kernel size of 3 and a standard deviation of 0.5 to smooth the cross-attention maps. For Stable Diffusion 3, the attention maps are extracted at a spatial resolution of $64 \times 64$ pixels, and we apply a Gaussian filter with a kernel size of 5 and a standard deviation of 0.5 to smooth the cross-attention maps.

Regarding the *alignment terms* in CTCAL, it is important to note that the magnitude of the cross-attention response for each token-specific attention map varies with the sampled timestep, whereas CTCAL primarily emphasizes spatial alignment. Therefore, we normalize each to the range [0, 1] to ensure consistency in scaling. Regarding the *regularization term* in CTCAL, our approach concentrates exclusively on attention maps corresponding to tokens that are semantically relevant to the given prompt. Specifically, we re-weight the attention values by excluding the specialized tokens *sot* and *eot*, and subsequently apply a softmax function over the remaining tokens.

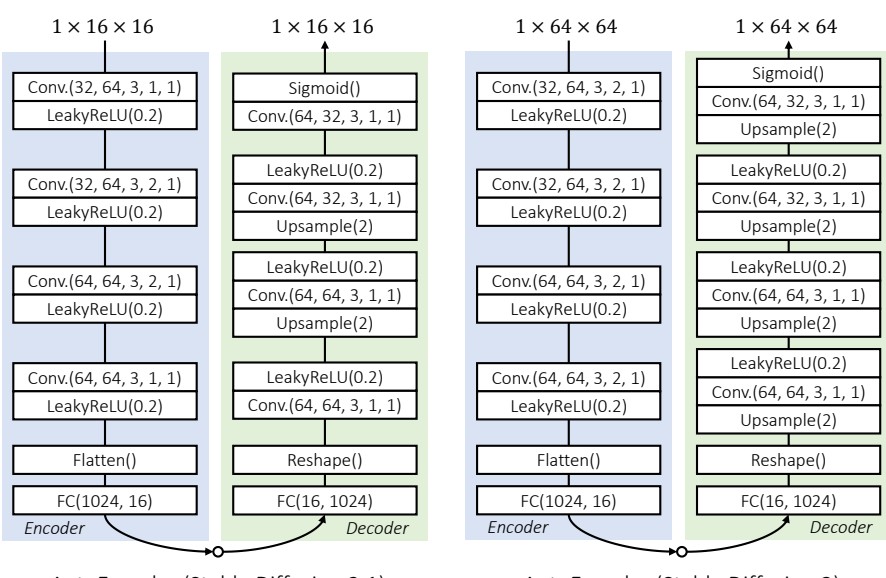

Figure 6: **The detailed architecture of the autoencoder.**

**Lightweight autoencoder.** The detailed architecture of the proposed lightweight autoencoder is illustrated in Fig. 6. We design dedicated autoencoders for Stable Diffusion 2.1 and Stable Diffusion 3, respectively, with the latter exhibiting a higher compression ratio. The latent code form extracted is the same in both cases.

**Training strategy for Stable Diffusion 3.** As outlined in the main paper, recent advancements in text-to-image diffusion models, such as Stable Diffusion 3, have shifted from traditional training paradigms that employ uniform timestep sampling to more sophisticated strategies incorporating non-uniform timestep samplers, notably logit-normal sampler (Esser et al., 2024). This methodolog-

ical evolution necessitates a reevaluation of timestep prioritization, particularly when determining $t_{\text{tea}}$.

Given timestep $t$, we define its priority as $w_t = \frac{p_t}{t}$, where $p_t$ represents the probability of sampling timestep $t$ under the employed distribution. Intuitively, timesteps characterized by lower noise levels (*i.e.*, smaller values of $t$) and higher sampling probabilities $p_t$ are assigned greater priority values $w_t$. Consequently, we select the timestep with the highest priority as $t_{\text{tea}}$.

We also provide the following PyTorch code for calculating $p_t$ in the case of a logit-normal sampler:

```python
from scipy.stats import norm

def logistic_interval_prob(a, b, mu, sigma):
    """
    Computes the probability that
        Y = sigmoid(X) falls within the interval [a, b],
        where X ~ N(mu, sigma^2).

    Parameters:
        a, b : Endpoints of the interval for Y (0 < a < b < 1).
        mu, sigma : Parameters of the Gaussian distribution.

    Returns:
        P(a <= Y <= b)
    """
    # Ensure numerical stability (avoid division by zero)
    a_clip = np.clip(a, 1e- 7, 1 - 1e- 7)
    b_clip = np.clip(b, 1e- 7, 1 - 1e- 7)

    # Compute the corresponding endpoints for X
    x_a = np.log(a_clip / (1 - a_clip))
    x_b = np.log(b_clip / (1 - b_clip))

    # Calculate the Gaussian CDF
    prob = norm.cdf(x_b, loc=mu, scale=sigma) - \
           norm.cdf(x_a, loc=mu, scale=sigma)
    return prob
```

## C  EXPERIMENTAL SETTINGS

**Training details.** Here we expand on training details and provide hyperparameters:

For SD 2.1, we utilize the AdamW optimizer, configuring the learning rate to $5 \times 10^{-5}$ for the denoising network and autoencoder, $5 \times 10^{-6}$ for the text encoder. The optimizer hyperparameters are set to $\beta_1 = 0.9$, $\beta_2 = 0.999$, and $\epsilon = 1 \times 10^{-8}$, with a weight decay of $0.01$. Training is conducted with a per-device batch size of 16 across 4 NVIDIA A800 GPUs, resulting in an effective batch size of 64. The fine-tuning is completed with 50000 training steps.

For SD 3, we similarly employ the AdamW optimizer. The denoising network and autoencoder are trained with a learning rate of $5 \times 10^{-5}$, while the text encoder utilizes a learning rate of $1 \times 10^{-5}$. The corresponding optimizer parameters are $\beta_1 = 0.9$, $\beta_2 = 0.999$, and $\epsilon = 1 \times 10^{-8}$, with a weight decay of $1 \times 10^{-4}$ for the denoising network and $1 \times 10^{-3}$ for the text encoder. Training is performed with a per-device batch size of 4 on 4 NVIDIA A800 GPUs, the gradient accumulation step is set to 4, yielding a total batch size of 64. Fine-tuning is performed for a total of 20,000 optimization steps.

**Tradeoff parameters.** CTCAL is a hyperparameter-robust method. Instead of meticulously tuning the tradeoff parameters, we empirically determine them by aligning the magnitudes of various loss terms. This straightforward strategy enables stable training convergence and performance gains. For SD 2.1, we empirically set $\lambda_1 = 0.01$, $\lambda_2 = 1$, $\lambda_3 = 0.01$, and $\lambda_4 = 0.1$. For SD 3, which we found to be less susceptible to the effects of imbalanced cross-attention responses among subjects on the T2I-CompBench++ benchmark, we empirically set $\lambda_1 = 0.01$, $\lambda_2 = 1$, $\lambda_3 = 0.01$, and $\lambda_4 = 0$.

For those seeking to further improve performance through extensive hyperparameter tuning, we suggest considering a strategy based on the gradient alignment of each loss term. Regarding concerns about performance degradation or training instability, we recommend appropriately increasing the weight of the diffusion loss to mitigate such issues.

**Benchmark.** We conduct a comprehensive evaluation of CTCAL utilizing two widely recognized benchmarks: T2I-CompBench++ (Huang et al., 2025) and GenEval (Ghosh et al., 2023). T2I-CompBench++ comprises 8,000 compositional prompts across eight categories, and serves to assess text-to-image generation models through a multifaceted evaluation framework, including visual question answering (VQA), object detection, image-text matching scores, and assessments by multimodal large language models (MLLMs). GenEval consists of 553 object-centric prompts and focuses on evaluating the compositional reasoning capabilities of text-to-image generation models, which utilizes object detection and color classification tasks to rigorously assess the ability to capture and reproduce fine-grained object properties as specified in the prompts.

**Baseline (GORS).** Generative mOdel finetuning with Reward-driven Sample selection (GORS) presents a straightforward yet effective methodology to enhance the compositional capabilities of pre-trained text-to-image diffusion models. The core idea is to fine-tune a pre-trained model, *e.g.*, Stable Diffusion (Rombach et al., 2022), using a curated set of generated images that exhibit a high degree of alignment with given text prompts, and the fine-tuning loss is modulated by a reward signal.

Formally, let $\theta$ denotes the text-to-image diffusion model and $\{\mathbf{y}_1, \mathbf{y}_2, \cdots, \mathbf{y}_n\}$ represents a collection of text prompts. For each prompt $\mathbf{y}_i$, GORS generates $k$ candidate images, yielding a total of $kn$ images $\{\mathbf{x}_1, \mathbf{x}_2, \cdots, \mathbf{x}_{kn}\}$. Each image $\mathbf{x}_j$ is assigned an alignment score $s_j$, serving as the reward metric. GORS then selects those images whose corresponding reward scores surpass a predefined threshold, forming a subset of samples $\mathcal{D}_s$ to be used for fine-tuning.

During the fine-tuning phase, the loss associated with each sample is weighted according to its reward score. The overall fine-tuning objective is expressed as:

$$\mathcal{L} = \mathbb{E}_{(\mathbf{x},\mathbf{y},s)\in\mathcal{D}_s} \left[ s \cdot \|\epsilon - \epsilon_\theta\left(\mathbf{x}, \mathbf{y}, t\right)\|_2^2 \right], \tag{9}$$

where $(\mathbf{x}, \mathbf{y}, s)$ is the triplet of the image, text prompt, and reward.

# D ADDITIONAL EXPERIMENTAL RESULTS

## D.1 MORE RESULTS ON DIVERSITY EVALUATION

To assess the diversity of images generated by the proposed method, we conduct a diversity evaluation experiment using the widely adopted Mean LPIPS Distance, where a higher value indicates greater diversity. As shown in Table 7, CTCAL improves text-image alignment performance without compromising the diversity of the generated samples.

Table 7: **More results on diversity evaluation.**

| Methods | Color Mean LPIPS Distance | 2D-Spatial Mean LPIPS Distance |
|---|---|---|
| SD 2.1 | 0.637 | 0.618 |
| SD 2.1 + GORS | 0.621 | 0.626 |
| SD 2.1 + CTCAL (Ours) | 0.634 | 0.623 |

## D.2 MORE RESULTS ON IMAGE QUALITY EVALUATION

We present additional quantitative experiments on aesthetic score, which is a reward function for measuring image quality that is independent of text alignment. As shown in the Table 8, our method does not compromise the quality of generated images while improving text-image consistency; on the contrary, it exhibits a moderate improvement in quality. This suggests that the advantage of CTCAL in text-image alignment is shared with image quality. Improved text-image alignment can

correct semantic confusion and conflicts in the spatial dimension, which enhances the ability to render the correct object in the accurate location, which in turn leads to improved image quality.

Table 8: **More results on image quality evaluation.**

| Methods | Color
Aesthetic score | 2D-Spatial
Aesthetic score |
|---|---|---|
| SD 2.1 | 5.128 | 5.263 |
| SD 2.1 + GORS | 5.194 | 5.281 |
| SD 2.1 + CTCAL (Ours) | **5.288** | **5.344** |

### D.3 MORE OBJECTIVE EVALUATION ON GENEVAL.

As an extension of Table 2 in the main paper, we show all objective evaluation results on GenEval in Table 9.

Table 9: **Objective evaluation on GenEval.** CTCAL improves performance across all categories.

| Methods | Overall | Single
object | Two
object | Counting | Colors | Position | Color
attribution |
|---|---|---|---|---|---|---|---|
| SD 1.5 (Rombach et al., 2022) | 0.43 | 0.97 | 0.38 | 0.35 | 0.76 | 0.04 | 0.06 |
| SD 2.1 (Rombach et al., 2022) | 0.50 | 0.98 | 0.51 | 0.44 | 0.85 | 0.07 | 0.17 |
| SD XL (Podell et al., 2024) | 0.55 | 0.98 | 0.74 | 0.39 | 0.85 | 0.15 | 0.23 |
| Pixart-$\alpha$ (Chen et al., 2024b) | 0.48 | 0.98 | 0.50 | 0.44 | 0.80 | 0.08 | 0.07 |
| Hunyuan-DiT (Li et al., 2024b) | 0.63 | 0.97 | 0.77 | 0.71 | 0.88 | 0.13 | 0.30 |
| DALL-E 2 (Ramesh et al., 2022) | 0.52 | 0.94 | 0.66 | 0.49 | 0.77 | 0.10 | 0.19 |
| DALL-E 3 (Betker et al., 2023) | 0.67 | 0.96 | 0.87 | 0.47 | 0.83 | 0.43 | 0.45 |
| FLUX.1-dev (Labs, 2024) | 0.67 | 0.99 | 0.81 | 0.79 | 0.74 | 0.20 | 0.47 |
| Sana (1.6B) (Xie et al., 2025) | 0.66 | 0.99 | 0.77 | 0.62 | 0.88 | 0.21 | 0.47 |
| SD 3 (2B) (Esser et al., 2024) | 0.62 | 0.98 | 0.74 | 0.63 | 0.67 | 0.34 | 0.36 |
| SD 3 (2B) + CTCAL | **0.69** | **0.99** | **0.85** | **0.70** | **0.79** | **0.38** | **0.42** |

### D.4 MORE APPLICATION ON INFERENCE ACCELERATION

The text-to-image diffusion model fine-tuned with CTCAL exhibits better consistency between cross-attention maps at larger timesteps and those at smaller timesteps. This phenomenon can be leveraged for inference acceleration.

To further validate this approach, we utilize TGATE (Liu et al., 2025), a simple and training-free inference acceleration method that efficiently generates images by caching and reusing attention outputs from the predefined timestep. Specifically, we adopt its design for cross-attention layers: during inference, cross-attention computation is performed only for the first $T_c$ timesteps, and the output from the $T_c$-th timestep is reused in subsequent inference steps.

The validation experiment is shown in Fig. 7, SD 2.1 fine-tuned with CTCAL significantly improves the quality of the generated images while showing excellent alignment with text prompts under smaller $T_c$ settings, for both hard samples (top) and simple samples (bottom).

### D.5 ADDITIONAL QUALITATIVE COMPARISON

Fig. 8 presents examples of complex prompts, and Fig. 9 presents additional qualitative comparisons. It can be observed that our method generates semantically more plausible and photorealistic results than its counterparts, successfully capturing all input concepts.

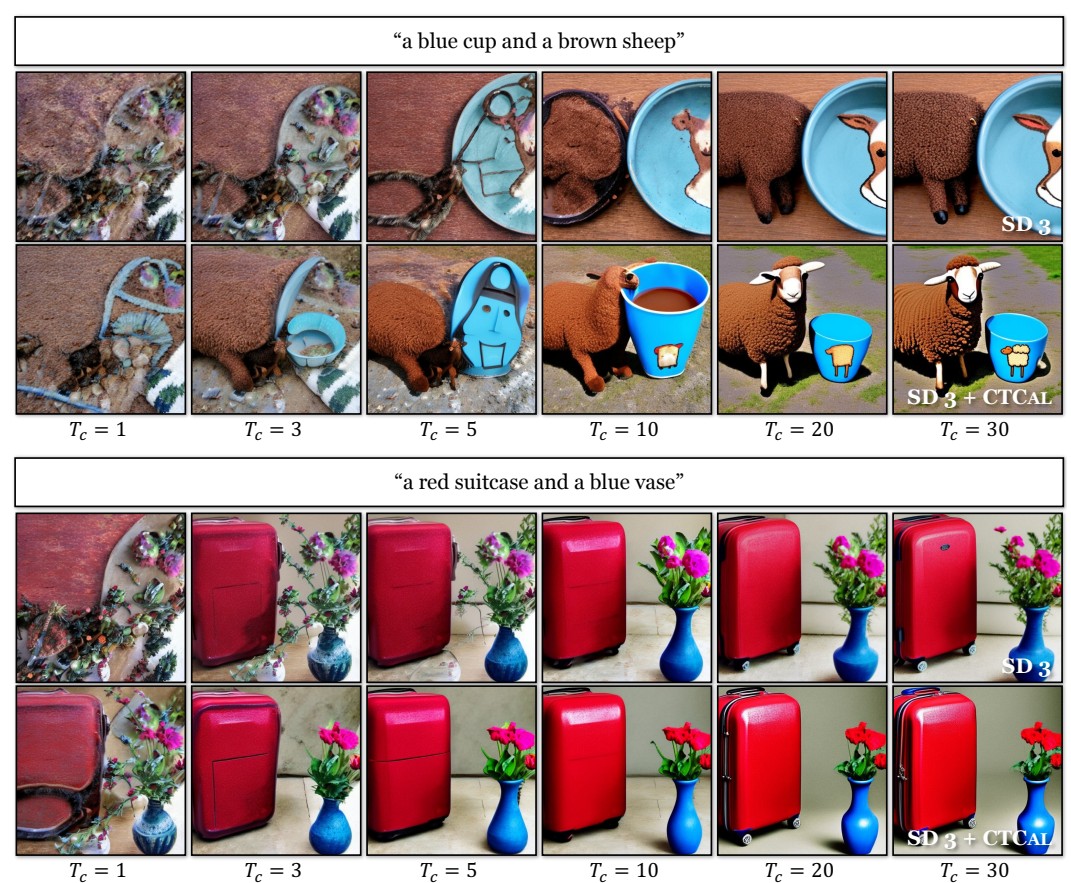

Figure 7: **More application on inference acceleration.**

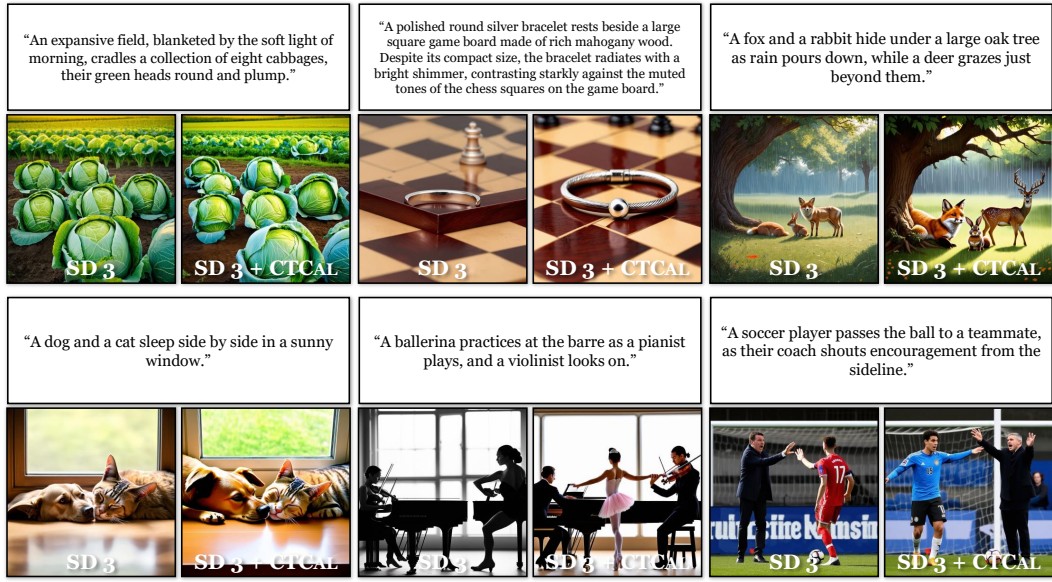

Figure 8: **More results** synthesized by the proposed CTCAL using complex text prompts.

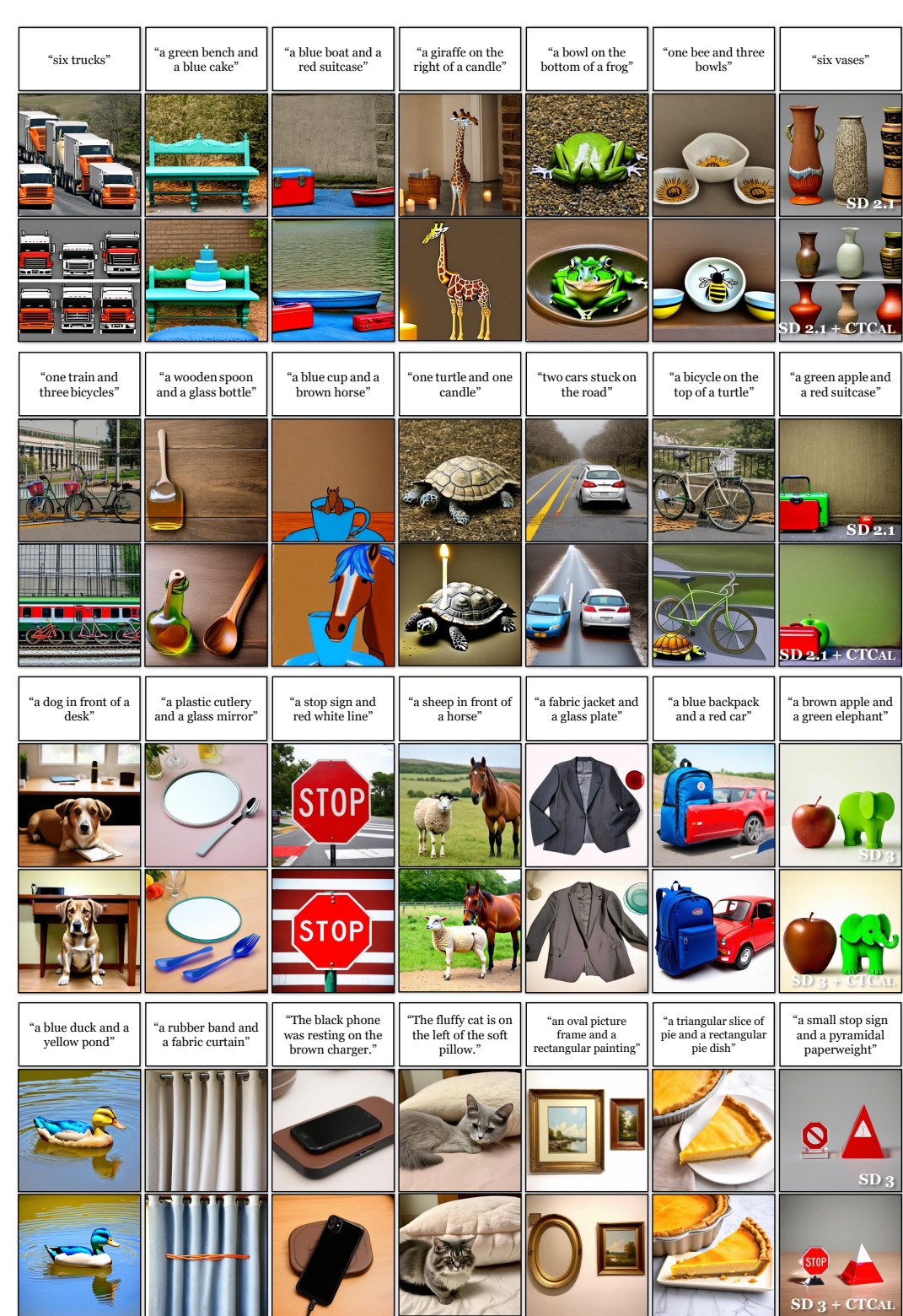

Figure 9: **More results** synthesized by the proposed CTCAL.

## E    LIMITATIONS

In this study, we utilize *Stanza* for part-of-speech analysis to extract nouns from text prompts, subsequently prioritizing the attention maps corresponding to these nouns for utilization in CTCAL. However, it is noteworthy that not every noun extracted via *Stanza* encapsulates meaningful spatial semantics. For instance, directional nouns such as *top* and *left* may be extracted; while a blacklist-based filtering mechanism can be employed to exclude such distractors, this method lacks generalizability and may not robustly address the issue across diverse prompts. The inadvertent inclusion of these nouns can consequently compromises the overall performance of CTCAL. To address this limitation, incorporating large language models to discern nouns that possess explicit physical spatial semantics emerges as a promising direction for enhancing the robustness and effectiveness of noun selection.

## F    LARGE LANGUAGE MODELS

Large Language Models are leveraged to refine and optimize the writing of this paper, performing grammatical error screening and correction.

