# OpenReview forum: "CTCal: Rethinking Text-to-Image Diffusion Models via Cross-Timestep Self-Calibration"
_ICLR.cc/2026/Conference — ICLR 2026 Conference Withdrawn Submission_

### Official Review · Reviewer_HxSy · 2025-10-30

**Soundness:** 2
**Presentation:** 2
**Contribution:** 2
**Rating:** 2
**Confidence:** 3

**Summary:**

This paper addresses the challenge of imprecise text-image alignment in text-to-image diffusion models, proposing Cross-Timestep Self-Calibration (CTCAL) based on the observation that alignment degrades with increasing (more noisy) timesteps .
CTCAL uses reliable cross-attention maps from smaller (less noisy) timesteps to calibrate larger-timestep learning, with components like noun-prioritized attention selection, pixel-semantic joint optimization, and subject response regularization, plus timestep-aware weighting for integrating with diffusion loss .
Model-agnostic CTCAL works with diffusion-based (e.g., Stable Diffusion 2.1) and flow-based (e.g., Stable Diffusion 3) models .
Experiments on T2I-CompBench++ and GenEval show CTCAL improves alignment in attributes, object relationships, and compositions, with user studies confirming better visual and semantic fidelity .

**Strengths:**

1. Targeting a validated issue: It addresses the measurable degradation of text-image alignment with increasing diffusion timesteps, supported by cross-attention map visualizations .

2. Model agnosticism: It seamlessly integrates into diverse text-to-image diffusion models, including diffusion-based (e.g., SD 2.1) and flow-based (e.g., SD 3) approaches .

3. Comprehensive validation: It is rigorously tested on T2I-CompBench++/GenEval benchmarks and user studies, with no trade-offs in image diversity or aesthetic quality .

**Weaknesses:**

1. Limited novelty. It is essentially an integration of existing techniques rather than a breakthrough: using cross-attention for alignment, filtering non-semantic tokens, and combining multi-loss terms are all well-explored in prior diffusion model optimization works. The token mapping in the attention map has been well-explored in previous works, either during inference process or training process. For example, , Dreamo[1] explores routing constraints in DiT structure to distinguish multiple subjects  and Anystory[2] explores multiple subjects injection on SDXL with attention maps restrictions.


2.  Fragile Noun Selection undermines core supervision. CTCAL’s performance depends on its POS-based filter, which relies on Stanza to extract "spatially meaningful" nouns. However, the paper admits this filter is flawed: it incorrectly includes non-spatial nouns (e.g., directional terms like "top" or "left") that lack semantic-spatial correspondence. The proposed workaround—an ad-hoc blacklist—is not generalizable, and the authors only mention "using LLMs for semantic filtering" as a future direction.

Reference:

[1] DreamO: A Unified Framework for Image Customization

[2]AnyStory: Towards Unified Single and Multiple Subject Personalization in Text-to-Image Generation

**Questions:**

please check the weaknesses.

---

> ### Author Response · Authors · 2025-11-13
>
> We sincerely appreciate the valuable feedback provided by the reviewer, and further thank the reviewer for the positive comments, including the model-agnostic nature and comprehensive validation of the proposed method.
>
> We found that the reviewer's main concerns stem from a misunderstanding of this work's contributions. It is **unobjective** for W1 to assess our contributions as an integration of existing technologies; furthermore, the finding of "token mapping in the attention map" mentioned in the comment is not our contribution. All these indicate that **our core contributions have been overlooked**. Furthermore, W2 constitutes an **implementation trade-off** and **does not undermine the core contributions** of this work. Thus, the provided review does not justify a score of 2.
>
> We address the reviewer's concerns in the following response:
>
> > **W1: Limited novelty.**
>
> **A1:** We thank the reviewer for pointing out two insightful works, but we must emphasize that they are **not strongly correlated** with our study. Furthermore, the contents highlighted by the reviewer, "using cross-attention for alignment, filtering non-semantic tokens, and combining multi-loss terms", **do not constitute the contribution of this work**.
>
> Our work focuses on the fine-tuning of text-to-image diffusion models, specifically dedicated to enhancing the performance of text-image alignment. We **identified the limitation in the conventional diffusion loss**, which fails to provide explicit supervision for modeling fine-grained text-image correspondences.
>
> **Proceeding from the timestep dimension**, and based on the phenomenon and supporting evidence observed during the pre-training phase: establishing accurate text-image alignment within diffusion models becomes progressively more difficult as the timestep increases. We propose CTCal, which leverages the reliable text-image alignment (i.e., cross-attention maps) formed at smaller timesteps with less noise to calibrate the representation learning at larger timesteps with more noise, thereby providing explicit supervision during training, thereby boosting text-image alignment performance. **Reviewer Ar73 accurately identified the contribution of this paper, recognizing it as a novel idea.**
>
> We also wish to emphasize that our method **does not require additional annotation tools or auxiliary models specifically for mask labeling** to facilitate the implementation of CTCal.
>
> Therefore, we kindly request the reviewer to re-evaluate the contributions of this paper.
>
> > **W2: Fragile Noun Selection undermines core supervision.**
>
> **A2:** We systematically discussed this issue in **Section E of the supplementary material of the original submission**. As the text prompts sourced from T2I-CompBench++ still qualify as simple text, employing the Stanza tool alongside a blacklist strategy represents a more cost-effective and sufficiently feasible solution. We also mentioned that for more complex prompts, LLM (Large Language Model) presents a viable, albeit higher-cost, yet still affordable alternative. **Therefore, this issue is merely an implementation trade-off and does not undermine the core contributions of this paper.**

---

> ### Author Response · Authors · 2025-11-14
>
> Dear Reviewer HxSy
>
> We sincerely apologize for the intrusion. The weaknesses you identified may primarily stem from your having inadvertently overlooked our core contribution, which **was recognized by Reviewer Ar73 as novel idea**.
>
> Since these issues may have affected your evaluation, could you consider revising your score upward if our response addresses your concerns adequately?
>
> Best regards,
>
> The authors of Paper #13367

---

### Official Review · Reviewer_Ar73 · 2025-10-31

**Soundness:** 2
**Presentation:** 3
**Contribution:** 2
**Rating:** 6
**Confidence:** 4

**Summary:**

This paper introduces a  fine-tuning method that leverages the text-image alignment (cross-attention maps) formed at smaller timesteps to explicitly calibrate the learning at larger timesteps. The goal of the paper is to improve the persistent challenge of poor alignment between the text prompt and generated images.

**Strengths:**

1. While a lot of prior works focus on improving image-text alignment during inference, its interesting to see this paper talk about providing explicit supervision for modeling fine-grained text-image correspondence during training instead.
2. The main idea of Cross-Timestep Self-Calibration is novel. It moves beyond conventional losses by introducing a self-supervised signal derived internally from the model's behavior at different levels of noise.

**Weaknesses:**

1. The method seems to add computational complexity, and the qualitative results do not seem strong enough to suggest utility of the proposed approach. For example, in the first half of Figure 4, the jar in the 5th column is just floating in the air, the banana in the 6th column looks unnatural and there is leakage of green to the banana.
2. The authors choose $t_{tea}=0$ in the final setup, but used t_{tea}=1 while motivating the overall approach in figure 1. I wonder whether t_{tea}=0 would give meaningful differentiation across attention maps for various objects unless I may be missing something.

**Questions:**

1. How reliable is the Part-of-Speech Tagger for complex prompts?
2. The authors mention in L203-204 that nouns (denoting objects or entities) are extracted and their attention maps are considered. I wonder whether the attributes of objects e.g. "yellow" for a yellow should be considered as well.
3. The real role of the autoencoder introduced is unclear to me. While a reconstruction task is used to prevent mode collapse, why would the alignment in the compressed space is better than pure pixel-level alignment?

---

> ### Author Response · Authors · 2025-11-13
>
> We sincerely appreciate the valuable feedback provided by the reviewer, and further thank the reviewer for the positive comments, particularly concerning the novelty of the ideas. We address the reviewer's concerns in the following response:
>
>
> > **W1: The method seems to add computational complexity, and the qualitative results do not seem strong enough to suggest utility of the proposed approach. For example, in the first half of Figure 4, the jar in the 5th column is just floating in the air, the banana in the 6th column looks unnatural and there is leakage of green to the banana.**
>
> **A1:** We thank the reviewer for the meticulous assessment of our qualitative results. We believe **these issues primarily stem from biases introduced by the training data**. As noted in Line 348 of the main paper, the research community currently lacks an open-source, high-quality text-image pair dataset specifically focused on text-image alignment. Consequently, we constructed our training dataset using synthetic data. The utilization of synthetic data inevitably introduces unrealistic scenarios, such as "floating jars." Such scenes are scarce in the real world, and synthesizing relevant data for them is also non-trivial.
>
> Furthermore, we wish to emphasize that, in these challenging scenarios, our method achieves a **significant improvement from failure to success (e.g., generating "glass jar" and "blue banana")** when compared to SD 2.1 and SD 2.1 + GORS. We consider this improvement sufficiently noteworthy.
>
> Naturally, we fully agree with the reviewer's concern regarding the need for further enhancement in image visual quality, and we believe utilizing high-quality real-world data may be a promising direction.
>
> > **W2: The authors choose $t_{tea}=0$ in the final setup, but used $t_{tea}=1$ while motivating the overall approach in figure 1. I wonder whether t_{tea}=0 would give meaningful differentiation across attention maps for various objects unless I may be missing something.**
>
> **A2:** We sincerely apologize for the misunderstanding caused to the reviewer. $t_\text{tea}$ corresponds to the state with minimal noise, not the clean image. Generally, the community also denotes the clean image as $I_0$, which may have contributed to the reviewer's confusion.
>
> > **Q1: How reliable is the Part-of-Speech Tagger for complex prompts?**
>
> **A3:** We systematically discussed this in **Section E of the supplementary material of the original submission**. Since the text prompts obtained from T2I-CompBench++ are still considered simple, using Stanza is a more cost-effective and perfectly viable solution. We also mentioned that for more complex prompts, LLM offers a viable, albeit more expensive, alternative that is still acceptable, as we only require a one-time pre-processing of the dataset.
>
> > **Q2: The authors mention in L203-204 that nouns (denoting objects or entities) are extracted and their attention maps are considered. I wonder whether the attributes of objects e.g. "yellow" for a yellow should be considered as well.**
>
> **A4:** We have provided additional experiments concerning adjectives (color and texture) in **Section 4.5 of the main paper (Line 456) of the original submission**. As shown in Table 6, the results indicate that incorporating alignment on adjective tokens leads to measurable improvements in performance.
>
> > **Q3: The real role of the autoencoder introduced is unclear to me. While a reconstruction task is used to prevent mode collapse, why would the alignment in the compressed space is better than pure pixel-level alignment?**
>
> **A5:** The alignments in the compressed space and the pixel space are complementary, as demonstrated by (B) and (C) in Table 4, where (C) incorporates compressed space alignment on top of the pixel space alignment utilized in (B). The compressed space and pixel space alignments respectively enforce global and local alignment constraints, constituting a classic complementary pair of constraints in image generation tasks.

---

> ### Author Response · Authors · 2025-11-14
>
> Dear Reviewer Ar73
>
> We sincerely apologize for the intrusion. As the weaknesses you identified mainly stem from the ambiguity in the **presentation of the main paper**, and nearly all the questions you raised have corresponding discussions and experiments **provided in the supplementary materials of the original submission**, we have promptly completed our response.
>
> Since these issues may have affected your evaluation, could you consider revising your score upward if our response addresses your concerns adequately?
>
> Best regards,
>
> The authors of Paper #13367

---

### Official Review · Reviewer_TZMs · 2025-11-01

**Soundness:** 2
**Presentation:** 3
**Contribution:** 2
**Rating:** 4
**Confidence:** 4

**Summary:**

The paper tackles the gap that text–image correspondence tends to be strong at small timesteps (low noise) but weak at large timesteps. It proposes a training-time self-calibration scheme (CTCAL): for each sample, a "teacher" small-timestep cross-attention map supervises the "student" large-timestep map. The loss combines pixel-level and semantic-level attention alignment, a subject-response balancing term, and a timestep-aware weighting that emphasizes harder (noisier) steps. On SD 2.1 and SD 3, the method improves compositional alignment and prompt faithfulness on standard benchmarks.

**Strengths:**

1. **Clear problem framing:** Directly targets cross-timestep misalignment.
2. **Simple supervision signal:** Reuses model-internal cross-attention as a self-supervised ``teacher".
3. **Comprehensive design:** Pixel + semantic attention alignment and subject balancing are sensible; timestep-aware weighting matches the difficulty profile.
4. **Empirical gains:** Consistent improvements on compositional/prompt-following metrics.

**Weaknesses:**

1. **Diversity risk from attention supervision.**
   Using small-timestep attention to shape large-timestep behavior might bias the model toward more deterministic layouts and reduce output diversity. I wonder is there a drop on diversity or mode collapse in generation. An metric analysis or visualization result may help.

2. **Dataset construction and generalization.**
   Training data is curated from T2I-CompBench-like prompts via reward-driven selection. This raises concerns about overfitting to that prompt style or metric. Please evaluate on broader benchmarks or metrics (e.g., FID, CLIPScore, HPS, ImageReward) to demonstrate the generalization.

3. **Positioning vs reward-based post-training (ReFL / GRPO family).**
   While CTCAL focuses on improving cross-timestep consistency during training, recent post-training methods such as ReFL or GRPO also enhance text–image alignment by optimizing explicit reward signals. It would strengthen the paper to clarify how CTCAL compares or complements these approaches—both conceptually and empirically. A short discussion or a compute-matched comparison (e.g., ReFL vs. CTCAL under similar reward setups) would help readers understand whether CTCAL offers distinct benefits or can be combined with reward-based fine-tuning.

4. **Complexity and scalability of the training recipe.**
   The approach combines several components (noun-focused maps, pixel+semantic alignment, subject regularization, timestep-aware weighting). It’s not obvious how robust this recipe is when scaling to larger/faster backbones. More evidence of training stability and a brief report of training cost or efficiency would make the method’s practicality more convincing.

**Questions:**

Please check the weaknesses.

---

> ### Author Response · Authors · 2025-11-13
>
> We sincerely appreciate the valuable feedback provided by the reviewer, and further thank the reviewer for the positive comments, including the problem identification, methodology design, and improved experimental results.
>
> We found that the reviewers' concerns primarily stemmed from **overlooking the discussion and experiments provided in the supplementary material of the original submission**. We address the reviewer's concerns in the following response:
>
> > **W1: Diversity risk from attention supervision.**
>
> **A1:** We fully agree with the reviewer's concern regarding diversity. In fact, in **Section D.1 of the supplementary material in our original submission (prior to the rebuttal)**, we provided a corresponding discussion and quantitative evaluation demonstrating that our method does not cause a reduction in diversity or mode collapse (mentioned by Reviewer HxSy). It is possible that the reviewer overlooked this crucial discussion because we did not explicitly point to it in the main paper.
>
> > **W2: Dataset construction and generalization.**
>
> **A2:** We fully acknowledge the reviewer's concerns regarding generalizability. In fact, in our **original manuscript submission (prior to the rebuttal)**, we provided corresponding discussions on this matter:
>
> 1. **Experiments on the GenEval Benchmark:** As mentioned in **Section 4.3 (Line 398)**, we provided a cross-benchmark validation. SD 3 fine-tuned with CTCal on the dataset derived from T2I-CompBench++ demonstrated improved performance on the GenEval benchmark. This indicates that our method does not cause overfitting to the T2I-CompBench++ benchmark, including the text prompt styles and metrics.
> 2. **Qualitative Experiments on Complex Prompts:** As illustrated in **Figure 8 of the supplementary material of the original submission**, our method still enhances performance on complex prompts, even though the training data was based on simple prompts from the T2I-CompBench++ benchmark. This demonstrates that the improvement in text-image alignment achieved by CTCal is generalizable.
> 3. **Regarding Additional Metrics:** We appreciate the reviewer's suggestion to include more metrics, such as FID, CLIPScore, HPS, and ImageReward. Unfortunately, these specific metrics perform poorly in evaluating text-image alignment. Therefore, this paper selected the more comprehensive and specialized T2I-Compbench++ and GenEval benchmarks, which are more appropriate for this evaluation.
>
> > **W3: Positioning vs reward-based post-training (ReFL / GRPO family).**
>
> **A3:** The training process of large models generally comprises three stages: pre-training, supervised fine-tuning (SFT), and reinforcement learning-based fine-tuning. CTCal falls into the supervised fine-tuning category, whereas ReFL and GRPO belong to the reinforcement learning category.
>
> CTCal is a fine-tuning method oriented toward high-quality data, while ReFL and GRPO are fine-tuning methods oriented toward reward models. As mentioned in **Section 4.1 of the main paper**, the research community lacks open-source, high-quality text-image pair datasets. Therefore, we adopt the dataset construction method proposed by (Huang et al., 2025), which utilizes a reward-driven sample selection strategy to curate the training dataset.
>
> However, it is crucial to note that **CTCal is not a reward-model-dependent fine-tuning method**. In scenarios where a reward model is absent, but a high-quality dataset is available, CTCal remains applicable, whereas ReFL and GRPO cannot be. CTCal is designed to better leverage high-quality text-image data for supervised fine-tuning; consequently, CTCal is **not in a strictly competitive relationship** with ReFL and GRPO. From the perspective of large model training paradigms, they represent two **complementary** phases.
>
> > **W4: Complexity and scalability of the training recipe.**
>
> **A4:** We discuss the hyperparameter selection and potential extensions of the proposed training scheme in the "Tradeoff parameters" subsection of **Section C in the supplementary material  of the original submission**. Our method demonstrates hyperparameter stability while consistently achieving performance improvements across different architectures. The training cost and details are also provided in the "Training details" subsection of **Section C in the supplementary material of the original submission**.

---

> ### Author Response · Authors · 2025-11-14
>
> Dear Reviewer TZMs
>
> We sincerely apologize for the intrusion. Most of the weaknesses you pointed out **have been addressed with corresponding discussions and experiments in the supplementary materials of the original submission**, allowing us to complete the response promptly.
>
> Since these oversights may have affected your evaluation, would you be kind enough to reconsider and adjust your score if our response resolves the concerns you raised?
>
> Best regards,
>
> The authors of Paper #13367

---

### Note · Authors · 2025-11-14

**Comment:**

We sincerely appreciate the valuable feedback and comments provided by all reviewers. Unfortunately, however, due to the reviewers' oversight of some important experiments and discussions in the supplementary materials, our work has received an unjustifiably low score.

We sought to engage in proactive and timely discussions with the reviewers, yet considering that such an approach might be inappropriate, we did not receive a timely response.

We express deep regret regarding this outcome and have decided to withdraw the manuscript. We would like to once again acknowledge the reviewers' contributions to the ICLR community.

**Withdrawal Confirmation:**

I have read and agree with the venue's withdrawal policy on behalf of myself and my co-authors.